# A Study on the Role of Intraoperative Corticobulbar Motor Evoked Potentials for Improving Safety of Cerebellopontine Angle Surgery in Elderly Patients

**DOI:** 10.3390/diagnostics13040710

**Published:** 2023-02-13

**Authors:** Quintino Giorgio D’Alessandris, Grazia Menna, Vito Stifano, Giuseppe Maria Della Pepa, Benedetta Burattini, Michele Di Domenico, Alessandro Izzo, Manuela D’Ercole, Liverana Lauretti, Nicola Montano, Alessandro Olivi

**Affiliations:** 1Department of Neurosurgery, Fondazione Policlinico Universitario A. Gemelli IRCCS, 00168 Rome, Italy; 2Department of Neuroscience, Neurosurgery Section, Fondazione Policlinico Universitario A. Gemelli IRCCS, Università Cattolica del Sacro Cuore, Largo Agostino Gemelli, 8, 00168 Rome, Italy

**Keywords:** elderly, cerebellopontine angle surgery, intraoperative neurophysiological monitoring, facial motor evoked potentials

## Abstract

Preservation of facial nerve function (FNF) during neurosurgery for cerebellopontine angle (CPA) tumors is paramount in elderly patients. Corticobulbar facial motor evoked potentials (FMEPs) allow assessment intraoperatively of the functional integrity of facial motor pathways, thus improving safety. We aimed to evaluate the significance of intraoperative FMEPs in patients 65 years and older. A retrospective cohort of 35 patients undergoing CPA tumors resection was reported; outcomes of patients aged 65–69 years vs. ≥70 years were compared. FMEPs were registered both from upper and lower face muscles, and amplitude ratios (minimum-to-baseline, MBR; final-to-baseline, FBR; and recovery value, FBR minus MBR) were calculated. Overall, 78.8% of patients had a good late (at 1 year) FNF, with no differences between age groups. In patients aged ≥70 years, MBR significantly correlated with late FNF. At receiver operating characteristics (ROC) analysis, in patients aged 65–69 years, FBR (with 50% cut-off value) could reliably predict late FNF. By contrast, in patients aged ≥70 years, the most accurate predictor of late FNF was MBR, with 12.5% cut-off. Thus, FMEPs are a valuable tool for improving safety in CPA surgery in elderly patients as well. Considering literature data, we noticed higher cut-off values for FBR and a role for MBR, which suggests an increased vulnerability of facial nerves in elderly patients compared to younger ones.

## 1. Introduction

Neurosurgery in the cerebellopontine angle (CPA) poses extraordinary technical challenges and requires advanced surgical skills, due to the crowding of key neuro-vascular structures in this compartment [1,2]. From an anatomical viewpoint, in fact, the cerebellopontine angle contains three neurovascular complexes in a relatively narrow space. In detail, the upper complex is built up by the superior cerebellar artery, the midbrain, the superior cerebellar peduncle, the tentorial surface of the cerebellum, and the oculomotor, trochlear, and trigeminal cranial nerves; the middle complex includes the antero-inferior cerebellar artery, the pons, the middle cerebellar peduncle, the petrosal surface of the cerebellum, and the abducens, facial, and vestibulocochlear nerves; and, finally, the lower complex comprises the postero-inferior cerebellar artery, the medulla oblungata, the inferior cerebellar peduncle, the suboccipital surface of the cerebellum, and the glossopharyngeal, vagus, spinal accessory, and hypoglossal nerves [3]. The most frequent neoplasms arising in this region are schwannoma and meningioma, with vestibular schwannomas largely representing the most common histotype [4]. Magnetic resonance imaging is crucial in establishing the correct diagnosis and in detailing the relationship between the tumor and the key neurovascular structures hosted in CPA [4]. T2 signal is typically high and non-homogeneous in schwannomas, while T1-dependent gadolinium enhancement is strong and homogeneous in meningiomas. Moreover, vestibular schwannomas are centered at the internal acoustic meatus and usually have an intracanalicular portion, while meningiomas growth is eccentric. Schwannomas arising from lower cranial nerves or from the trigeminal nerve are centered on the jugular foramen or on the Meckel cavum, respectively. Epidermoid cysts are rarer in CPA, and can be detected pre-operatively based on the marked restriction of diffusion in DWI images.

As already pointed out by many groups including ours [5], CPA tumors are generally resected using the retrosigmoid approach, which allows the maximizing of exposure while minimizing surgical hazards. In this surgery, safety preservation is paramount. Against this background, facial (FN) function (FNF) preservation is essential. It has been demonstrated that facial palsy negatively affects quality of life [6] and damage to this nerve is frequently associated with injuries to other structures, including lower cranial nerves, with possible detrimental effects also on swallowing and phonation [7,8]. Unfortunately, a reduced FNF has been reported in up to 40% of cases after neurosurgery for CPA tumors [6,9]. For these reasons, CPA surgery is reserved for young and “fit” patients, while elderly ones have been traditionally under-treated.

However, in the framework of increasing life expectancy and global population ageing, a higher number of frail and old patients seek surgical procedures [10,11,12,13,14]. Elderly neuro-oncological patients have recently generated growing interest in the neurosurgical community. As a result, a series of studies have been published on the topic. In most of these studies, the elderly were defined as being over 65 years old [10,15,16]. In this slice of the population, the issue of safety through a careful balancing of surgical resection and preservation of neurological function is even more crucial than in younger ones, given on the one hand, the reduced resilience, and on the other, the shorter life expectancy. 

We and other groups recently demonstrated that the intraoperative use of corticobulbar facial motor evoked potentials (FMEPs) represents a support to the well-known intraoperative neurophysiological monitoring (IONM) techniques, as direct electrical stimulation (DES) of the nerve and free-running electromyography (EMG), and can reliably predict early and late post-operative FNF in CPA surgery [17,18,19,20,21,22].

The aim of the present study was to assess the impact of intraoperative FMEPs on FNF in patients beyond 65 years of age undergoing CPA surgery. 

## 2. Materials and Methods

### 2.1. Patients Enrollment and Clinical Data

We retrospectively reviewed the clinical, neurophysiological and surgical data of 35 patients aged 65 years or older who underwent neurosurgery under IONM for resection of extra-axial tumors involving CPA between July 2018 and December 2021 at the Department of Neurosurgery, A. Gemelli University Hospital IRCCS, Rome, Italy. A subset of patients (n = 23) was included in a previously-published surgical series [19]. All patients signed an informed consent form according to the research purposes of the Institutional Ethics Committee of A. Gemelli University Hospital IRCCS. 

FNF was described using the House-Brackmann scale (HB), as follows: grade I, normal facial function; grade II, mild dysfunction; grade III, moderate dysfunction with normal symmetry at rest and complete eye closure with maximal effort; grade IV, moderately severe dysfunction, with disfiguring asymmetry and incomplete eye closure; grade V, severe dysfunction with barely perceptible motion; and grade VI, no movement. For the purposes of the present study, according to literature [23], FNF using the HB scale was dichotomized as good (HB I-II) or poor (HB III-VI). We excluded patients with poor FNF at admission. FNF at discharge (“early FNF”) and at 1-year follow-up (“late FNF”) was registered.

### 2.2. Intraoperative Neurophysiological Monitoring

Technique for IONM has been previously reported in detail [5,16,17,19,20,21,22,23,24,25,26,27]. To measure FMEPs, electrodes were placed at C3, C4, and Cz scalp location; to stimulate facial motor area contralateral to the tumor side, a cathode was placed in Cz and anode was placed in C3 or C4. Stimulation parameters were as follows: single pulse 350 to 500 μs; train of pulses 4 to 7 stimuli, 350 to 500 μs, 500 to 700 Hz, delivered at 40 ms from the first single pulse. A response was considered central only if generated after the train of pulses; vice versa, if a response was already present after single pulse stimulation, it was regarded as peripheral and deemed spurious.

From FMEP amplitudes, we calculated the minimum-to-baseline amplitude ratio (MBR), the final-to-baseline amplitude ratio (FBR), and the recovery value (RV = FBR minus MBR) [11,13]. These parameters were calculated from orbicularis oculi (OOc) for upper facial muscles, and orbicularis oris (OOr) for lower facial muscles. IONM setting was completed by DES, free-running EMG, and motor and somatosensory evoked potentials [17]. NIM Eclipse IONM system (Medtronic Inc., Minneapolis, MN, USA) was used. Neuronavigation was used in all cases [20]. 

### 2.3. Statistical Analysis

Descriptive analysis was performed using mean (±SD) or median (range), and absolute and relative frequencies for continuous and categorical variables, respectively. Chi-square test (using Fisher exact test when appropriate) was used for comparison of categorical variables, and Mann-Whitney *U* test for comparison of continuous variables between groups.

To analyze the accuracy of amplitude indices in predicting a good FNF, receiver operating characteristics curves (ROC) were plotted and the area under the curve (AUC) was calculated, using the Hanley and McNeil method. The best cut-off was defined as the value at which the Youden index, i.e., the difference between sensitivity and 1-specificity, had its maximum value. A *p* value < 0.05 was considered statistically significant for all analyses. Statistical analysis was performed using StaView ver 5.0 (SAS Institute, Cary, NC, USA) and MedCalc ver 20.015 (MedCalc Software Ltd., Ostend, Belgium).

## 3. Results

### 3.1. Patients Characteristics

Thirty-five patients were enrolled in the present study; 17 of them were aged 65–69 years, the remaining 18 were aged ≥70 years. Clinical features are presented in Table 1 and neurophysiological values in Table 2. No significant differences between early and late post-operative FNF were detected between the two groups. Schwannoma was the most frequent diagnosis in both groups, though it was more frequently recorded in the 65–69 years group (*p* = 0.03, Fisher Exact test); follow-up was longer in the ≥70 years group (*p* = 0.02, Mann-Whitney *U* test). 

### 3.2. Predictors of Early Post-Operative FNF

In the subgroup of patients aged 65–69 years, we could not find significant differences in the calculated FMEP indices between patients with good and poor early post-operative FNF (Figure 1). This notwithstanding, at ROC analysis, FBR as registered from OOc muscle was able to reliably predict early post-operative FNF (AUC = 0.857, *p* = 0.0062), with a cut-off value set at zero (Table 3 and Figure 2).

Conversely, in the subgroup of patients aged ≥70 years, both MBR and FBR were significantly higher in patients with good early post-operative FNF than in those with poor early post-operative FNF (Figure 1). At ROC analysis, FBR as registered from OOr muscle was the most reliable index (AUC = 0.937, *p* < 0.0001) for early post-operative FNF prediction, with a cut-off value set at 60% (Table 2 and Figure 2). 

### 3.3. Predictors of Late Post-Operative FNF 

In the subgroup of patients aged 65–69 years, no significant differences in FMEP indices were detected between patients with good and poor late post-operative FNF (Figure 3). At ROC analysis, FBR (registered from OOr) reliably predicted late post-operative FNF (AUC = 0.833, *p* = 0.0091), with a cut-off value set at 50% (Table 4 and Figure 2). 

In the subgroup of patients aged ≥70 years, only MBR (registered from OOr muscle) was significantly higher in patients with good vs. poor late post-operative FNF (Figure 3). MBR from OOr was also the most reliable predictor of good late post-operative FNF at ROC analysis (AUC = 0.931, *p* < 0.0001), with a cut-off value set at 12.5% (Table 4 and Figure 2).

## 4. Discussion

CPA cistern is a triangular CSF filled subarachnoid cistern lying between the anterior surface of the cerebellum and the lateral surface of the pons. It is bounded superiorly by the tentorium cerebelli, inferiorly by the lower cranial nerves, anteriorly by the prepontine cisterne, antero-laterally by the posterior surface of the petrous temporal bone, and medially by the pons. The main structures contained in this space are the seventh and eigth cranial nerves, floccolus of the cerebellum, foramen of Luschka of the fourth ventricle, and anterior inferior cerebellar artery. 

Given the high concentration of vital structures, CPA surgery requires special attention to patients’ safety, carefully balancing surgical radicality with preservation of neurological function.

Tumors of the CPA account for 5–10% of all intracranial neoplasms. Vestibular schwannoma and meningioma are the most frequent in the adult population, while medulloblastomas are among the most common primary brain tumors in children. Even if our series includes only schwannoma and meningioma, a wide range of pathologies can be actually seen in the CPA region. Based on MRI appearance, CPA masses can be classified into four groups: enhancing masses (vestibular Schwannoma, meningioma, trigeminal Schwannoma, facial nerve Schwannoma, ependymoma, metastasis), high T1 signal masses (hemorrhagic vestibular Schwannoma, neuroenteric cyst, thrombosed berry aneurysm white epidermoid, ruptured intracranial dermoid), CSF-like density masses (such as epidermoid cyst and arachnoide cyst), and other rarer lesions (primary melanocytc neoplasm, neurosarcoidosis, cholesterol granuloma, etc.). 

### 4.1. Safety of CPA Surgery in Elderly Patients

We and other groups have demonstrated that standard FNF monitoring techniques (DES and free-running EMG) only partially allow achieving the goal of functional preservation in CPA [10,15,16,17,24,25], while the integration of FMEP has the extraordinary potential to continuously monitor the functional integrity of facial motor pathway throughout the surgery [2,23,26,27,28,29,30,31]. 

A growing series of recently-published studies focused on the elderly neuro-oncological patients [10,15,16,17]. A milestone paper published by Chibbaro et al. [32] highlighted a progressive and significant increase in elderly presenting for neurosurgical elective surgeries over the last 25 years. However, the elderly are a heterogeneous group in terms of functional status, prognosis, and risk of adverse events in case of surgical procedures. Even if most of the treatment decisions are still heavily based on chronological age and the presence of comorbidities, there was a significant improvement in peri-operative standard-of-care due to innovations in surgical technique, overall greater understanding/knowledge of geriatric medicine, and upgrading in anaesthesiological techniques [33]. Against this background, several assessment tools have begun to be routinely used to assess the needs of older people and improve their long-term outcomes shortly after surgery. Among these we can list the comprehensive geriatric assessment (CGA), which is a robust but cumbersome assessment including six different domains (medical, mental health, functional capacity, social circumstances, environment, and risk score), and the clinical frailty score (CFS), focused on frailty, which is intended as a physiological state of increased vulnerability to stressors due to decreased physiological reserve [30]. Even if the use of these tools has been shown to overall improve post-operative outcomes, identification of kits specific for elderly neuro-oncological patients is necessary for them to be routinely used.

Using this approach, in this study we obtained a satisfactory long-term FNF outcome in elderly patients undergoing CPA surgery, with 78.8% of them achieving good late post-operative FNF. Notably, in patients aged ≥70 years, we obtained good long-term outcomes in 82.4% of cases. This evidence is reassuring on the safety and feasibility of CPA surgery in elderly patients, going against the general notion that this type of surgery is less safe for the elderly, with a significantly higher number of complications and a greater risk of recurrence.

### 4.2. Predictors of FNF in Elderlies

In addition, we assessed the possible predictive role of intraoperative FMEP on FNF outcomes. We had already addressed this issue in a cohort of 83 patients whose median age was 53 years [16]. In that paper, we found that FBR was able to reliably predict early and late FNF; moreover, we showed FMEPs followed an all-or-nothing rule, in which the preservation of some potential at the end of surgery (even if the original value was markedly reduced) was a good predictor of long-term facial recovery [11]. However, it is known that neural plasticity is overall impaired due to ageing [2,31]. The mechanism by which age negatively impacts on neuroregeneration is multifactorial and not fully understood [2,34,35,36]. Sprouting from α-motor axons is impaired with ageing, as well as the ability of α-motoneurons for axonal elongation. Moreover, the rate of synthesis and axonal transportation of cytoskeletal proteins is reduced, as well as the ability of nerve fibers to transport trophic factor. The capability of neurons to respond to stimuli by neurotrophic factors is also impaired, due to the reduction of expression of the dedicated receptors. Eventually, this is reflected by the impaired re-myelination ability of aging fibers. Finally, ageing has been associated with failure to restore a correct synaptic transmission at the neuromuscular junction [2]. Not negligibly, the reduced plasticity in the central nervous system also has a negative impact on the recovery ability of peripheral nerves [28]. In summary, there is a strong rationale supporting our experimental hypothesis, i.e., that data regarding the maintenance and recovery of FNF after CPA surgery, gathered on an unselected patients’ cohort, cannot be automatically translated to elderly patients. 

The results of this study seem to confirm this hypothesis. In the subgroup of patients aged 65–69 years, the most reliable predictor of post-operative FNF was FBR. This implies that transient intraoperative reductions of FMEP amplitudes do not have a detrimental impact on outcome, provided they recover at the end of surgery. However, for late post-operative FNF, the all-or-nothing rule seems invalid in this particular subset of patients, since the cut-off value was set at 50% (Figure 2); this means that FMEP amplitude should not fall to more than half the initial value to guarantee recovery. To summarize, in this population of “young elderly” patients, facial nerves appear more susceptible to damage than in the general population. 

In the subgroup of patients aged ≥70 years, this latter behavior appears magnified. Early outcome was still predicted reliably by FBR, but with a higher cut-off (60%; Figure 2). The most accurate predictive factor for late post-operative FNF was MBR. In other words, in patients aged ≥70 years undergoing CPA surgery, a substantial intraoperative reduction of FMEP amplitude, although transient, impairs the long-term functional outcome. This confirms our assumption that, in elderly patients, FN is definitely more susceptible to injury than in younger ones.

### 4.3. Potential Implications for Neurosurgery in Other Brain Areas

The widespread use of intraoperative neuromonitoring techniques is expected to broaden the neurosurgical indications in elderly patients. Such an assumption holds true particularly for tumors located in eloquent brain areas, such as motor and speech ones, and for medullary tumors. Appropriate cut-offs and settings for IONM, brain mapping and D-wave recording shall be adapted to elderly patients during these surgeries. Notably, the integration with advanced neuroradiology techniques, such as resting-state functional MRI and tractography in computed neuronavigation systems, will foster the safety of surgery. Resection of tumors located in proximity to language areas and bundles poses a major hurdle, since awake surgery routinely used in younger patients cannot be adapted to elderly ones. Further advancements in IONM, including cortico-cortical evoked potentials [37], could help in this respect.

### 4.4. Limitations

Patients have been homogeneously treated and followed up in a national neuro-oncologic referral center, reassuring on the quality of gathered data. However, the results of our study are not devoid of limitations, including: -the retrospective nature of data collection;-a possible selection bias, since elderly patients with major contraindications to general anesthesia were a priori excluded; and-the relatively low number of patients, which reduces the statistical significance.

## 5. Conclusions

Our study confirms that CPA surgery is safe and feasible in elderly patients. FMEPs are able to reliably predict post-operative FNF in this cohort. Notably, our work shows that ageing negatively impacts on facial nerve resistance; the results of our analysis, in fact, show that ability to recover after transient damage is reduced in elderly patients compared to younger ones. To conclude, in elderly patients, special attention must be paid to minimize FN manipulation. However, further studies, better if multicentric, are needed to confirm the generalizability of our results, and the ultimate role of FNF monitoring in this subset of the population.

## Figures and Tables

**Figure 1 diagnostics-13-00710-f001:**
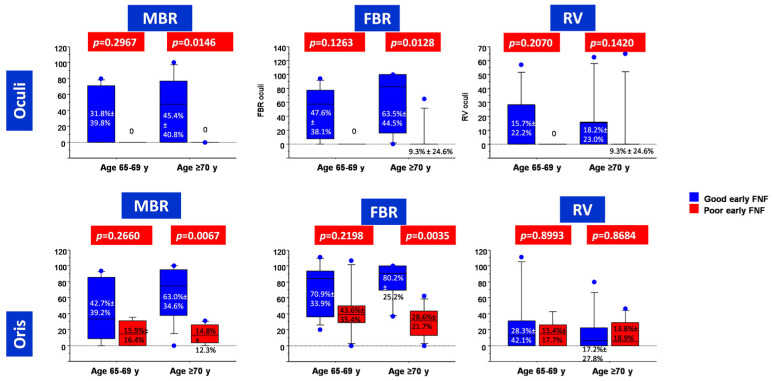
Box plots depicting corticobulbar FMEP indices depending on age and early post-operative FNF.

**Figure 2 diagnostics-13-00710-f002:**
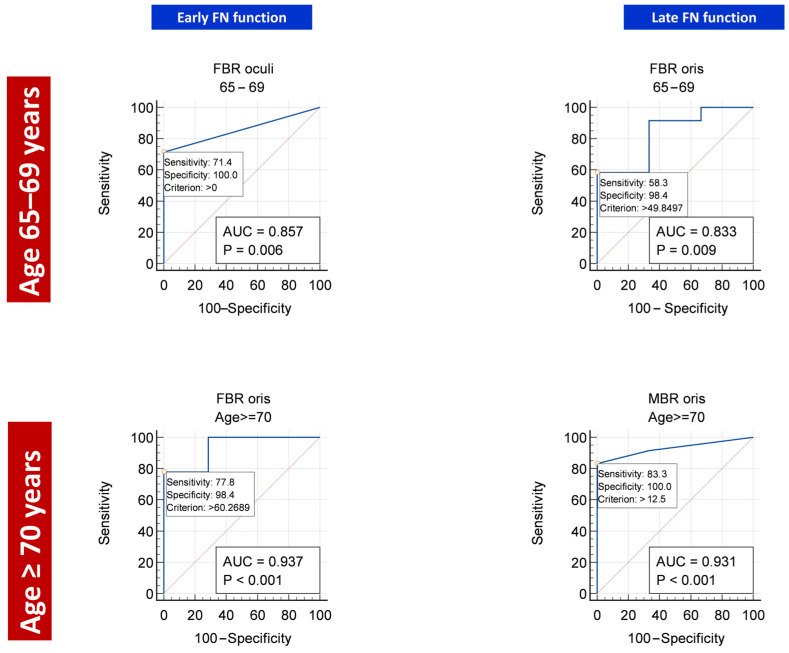
Best ROC curves for early (**left column**) and late FNF (**right column**) in patients aged 65–69 years (**upper row**) and ≥70 years (**lower row**).

**Figure 3 diagnostics-13-00710-f003:**
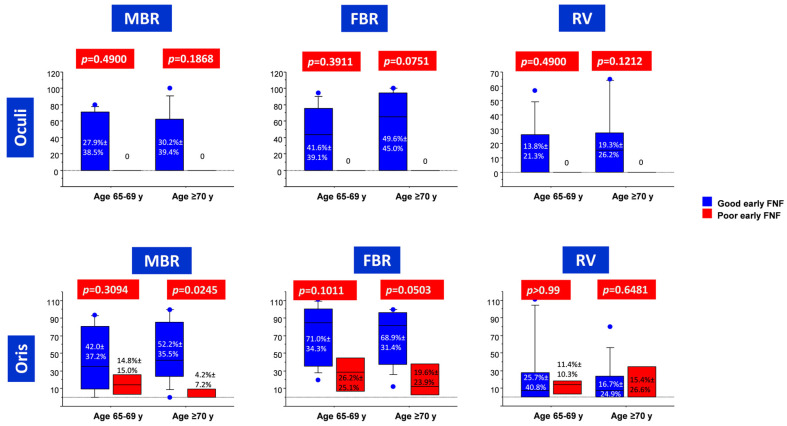
Box plots depicting corticobulbar FMEP indices depending on age and late post-operative FNF.

**Table 1 diagnostics-13-00710-t001:** Patients’ Characteristics.

Parameter	Value (Age 65–69 Years)	Value (Age ≥ 70 Years)	*p*
*n* patients	17	18	NA
Age (mean ± SD)	66.8 ± 1.5 years	73.4 ± 2.8 years	NA
Sex (M:F, *n* (%))	7:10 (41.2%:58.8%)	7:11 (38.9%:61.1%)	<0.99 ^#^
Pathology Schwannoma Meningioma	13 (76.5%) 4 (23.5%)	10 (55.6%) 8 (44.4%)	0.0315 ^#^
Follow-up (median, range)	12 months (6–41)	13.5 months (6–33)	0.0244 ^§^
GTR, *n* (%)	11 (64.7%)	16 (88.9%)	0.1212 ^#^
Good FN function (House-Brackmann 1–2) Pre-operative Early post-operative Late post-operative *	17 (100%) 10 (58.8%) 12 (75%)	18 (100%) 10 (55.6%) 14 (82.4%)	>0.99 ^#^ >0.99 ^#^ 0.6880 ^#^

* Late post-operative follow-up was available for 16 patients (age 65–69 years) and for 17 patients (age ≥ 70 years). ^#^ Fisher Exact Test; ^§^ Mann-Whitney *U* test.

**Table 2 diagnostics-13-00710-t002:** Corticobulbar facial motor evoked potentials (FMEP) indices.

Parameter	Age 65–70 Years	Age ≥ 70 Years	*p ^§^*
Minimum-to-baseline amplitude ratio (MBR) (mean ± SD) Oculi Oris	24.8% ± 37.2% 33.7% ± 35.1%	20.9% ± 35.3% 41.9% ± 36.2%	0.84 0.4379
Final-to-baseline amplitude ratio (FBR) (mean ± SD) Oculi Oris	37.0% ± 39.1% 60.8% ± 35.9%	34.3% ± 43.8% 57.6% ± 35.0%	0.9147 0.7343
Recovery Value (RV) (mean ± SD) Oculi Oris	11.0% ± 19.7% 24.0% ± 35.6%	13.4% ± 23.3% 15.7% ± 23.6%	0.7705 0.5379

^§^, Mann-Whitney *U* test.

**Table 3 diagnostics-13-00710-t003:** Area under the curve (AUC) of ROC curves analyzing the accuracy of FMEP indices in predicting good early post-operative FN function.

FMEP Index (Age)		Age 65–69 Years		Age ≥ 70 Years	
		AUC	*p*-Value	AUC	*p*-Value
Orbicularis oculi	MBR	0.714	0.2453	0.833	0.0091
	FBR	0.857	0.0062	0.881	0.0003
	RV	0.714	0.1992	0.714	0.1725
Orbicularis oris	MBR	0.680	0.2024	0.905	<0.0001
	FBR	0.700	0.1513	0.937	<0.0001
	RV	0.520	0.8999	0.524	0.8733

**Table 4 diagnostics-13-00710-t004:** Area under the curve (AUC) of the different ROC curves plotted to analyze the accuracy of the different FMEP indices from orbicularis oculi and oris (*left columns*) in predicting good late post-operative FN function in the two age groups.

FMEP Index		Age 65–69 Years		Age ≥ 70 Years	
		AUC	*p*-Value	AUC	*p*-Value
Orbicularis oculi	MBR	NA	NA	0.722	0.1433
	FBR	NA	NA	0.833	0.0049
	RV	NA	NA	0.778	0.0416
Orbicularis oris	MBR	0.697	0.1803	0.931	<0.0001
	FBR	0.833	0.0091	0.875	0.0005
	RV	0.500	>0.99	0.583	0.6780

## Data Availability

Data are available upon reasonable request to the corresponding author, N.M.

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
