# Peer review of "A Study on the Role of Intraoperative Corticobulbar Motor Evoked Potentials for Improving Safety of Cerebellopontine Angle Surgery in Elderly Patients"

_diagnostics, 2023, doi:10.3390/diagnostics13040710_

Round 1

Reviewer 1 Report

Quintino Giorgio D’Alessandris and co-workers submitted the paper entitled “A study on the role of intraoperative corticobulbar motor evoked potentials for improving safety of cerebellopontine angle surgery in elderly patients” for the consideration to publish in “Diagnostics (I.F= 3.992)”. This manuscript is based on safety relevant to neurosurgery with certain statistical analysis. Overall, this study looks fine, but some issues should be fixed as noted below.

1. In the introduction justify the selection of elderly patients above 65 years. Because, patients above 60 years are known as elderly peoples. Why they avoid 61 – 64 years age group?

2. In the abstract define ROC (receiver operating characteristics curves) while mentioning it first time.

3. In Figures 1 and 3, enhance the X and Y-axis font size and inner text font size for the readers.

4. Improve the discussion on 4.1 Safety of CPA surgery in elderly patients and deliver 4.3 Limitations in bullet points.

5. Conclusion section must be improved with merits and future studies.

Author Response

Quintino Giorgio D’Alessandris and co-workers submitted the paper entitled “A study on the role of intraoperative corticobulbar motor evoked potentials for improving safety of cerebellopontine angle surgery in elderly patients” for the consideration to publish in “Diagnostics (I.F= 3.992)”. This manuscript is based on safety relevant to neurosurgery with certain statistical analysis. Overall, this study looks fine, but some issues should be fixed as noted below.

  1. In the introduction justify the selection of elderly patients above 65 years. Because, patients above 60 years are known as elderly peoples. Why they avoid 61 – 64 years age group?

Response: We Thank the reviewer for this comment. Elderly neuro-oncological patients have recently generated growing interest in neurosurgical community. As a results, a series of studies were published on the topic. In most of these studies, elderly  were defined as over 65 years old. [Schär RT, Tashi S, Branca M, et al. How safe are elective craniotomies in elderly patients in neurosurgery today? A prospective cohort study of 1452 consecutive cases. J Neurosurg. 2020;134(3):1113-1121. Published 2020 Apr 24. doi:10.3171/2020.2.JNS193460; Klingenschmid J, Krigers A, Kerschbaumer J, Thomé C, Pinggera D, Freyschlag CF. Surgical Management of Malignant Glioma in the Elderly. Front Oncol. 2022;12:900382. Published 2022 May 26. doi:10.3389/fonc.2022.900382; Ferroli P, Vetrano IG, Schiavolin S, et al. Brain Tumor Resection in Elderly Patients: Potential Factors of Postoperative Worsening in a Predictive Outcome Model. Cancers (Basel). 2021;13(10):2320. Published 2021 May 12. doi:10.3390/cancers13102320]. These evidences justify our choice.

  1. In the abstract define ROC (receiver operating characteristics curves) while mentioning it first time.

Response: Done

  1. In Figures 1 and 3, enhance the X and Y-axis font size and inner text font size for the readers.

Response: Done.

  1. Improve the discussion on 4.1 Safety of CPA surgery in elderly patients and deliver 4.3 Limitations in bullet points.

Response: Discussion was improved in paragraph 4.1 and Limitations were reported in bullet points, as suggested.

  1. Conclusion section must be improved with merits and future studies.

Response: Done

Reviewer 2 Report

The submission " A study on the role of intraoperative corticobulbar motor evoked potentials for improving safety of cerebellopontine angle surgery in elderly patients" presents a study to estimate the impact of intraoperative facial motor evoked potentials on facial nerve function in elderly patients. The topic is interesting and this study can serve more in advance individuals. The paper is well-written and organized. The paper's methodology makes a scientific sound. However, some notices should be addressed to improve the manuscript for the publication process.

·         The similarity is high (close to 29%); it should be improved.

·         It is better if the author adds more previous state-of-the-art studies in the introduction section.

·         In the statistical analysis section, how the receiver operating characteristics curves (ROC) and the area under the curve (AUC) are calculated? The author needs to provide the required equations.

·         Table 4 requires more description.

Author Response

The submission " A study on the role of intraoperative corticobulbar motor evoked potentials for improving safety of cerebellopontine angle surgery in elderly patients" presents a study to estimate the impact of intraoperative facial motor evoked potentials on facial nerve function in elderly patients. The topic is interesting and this study can serve more in advance individuals. The paper is well-written and organized. The paper's methodology makes a scientific sound. However, some notices should be addressed to improve the manuscript for the publication process.

  • The similarity is high (close to 29%); it should be improved.

Response: Done

  • It is better if the author adds more previous state-of-the-art studies in the introduction section.

Response: Referees to previous state of the art topics were addended to the introduction section.

  • In the statistical analysis section, how the receiver operating characteristics curves (ROC) and the area under the curve (AUC) are calculated? The author needs to provide the required equations.

Response: as per standard method, the ROC curve was drawn by plotting the true positive rate (sensitivity) in function of the false positive rate (100-Specificity) for different cut-off points of a parameter. Using the MedCalc ver 20.015 software, AUC was estimated using the Hanley&McNeil method, as specified in the Methods section.

  • Table 4 requires more description.

Response: we have improved the description of Table 4.